# Direct laser writing of 3D electrodes on flexible substrates

Morgan A. Brown[1,3], Kara M. Zappitelli[1,3], Loveprit Singh[1], Rachel C. Yuan[1], Melissa Bemrose [1], Valerie Brogden[1], David J. Miller[1], Matthew C. Smear [1], Stuart F. Cogan[2] & Timothy J. Gardner [1] ✉

This report describes a 3D microelectrode array integrated on a thin-film flexible cable for neural recording in small animals. The fabrication process combines traditional silicon thin-film processing techniques and direct laser writing of 3D structures at micron resolution via two-photon lithography. Direct laser-writing of 3D-printed electrodes has been described before, but this report is the first to provide a method for producing high-aspect-ratio structures. One prototype, a 16-channel array with 300 μm pitch, demonstrates successful electrophysiological signal capture from bird and mouse brains. Additional devices include 90 μm pitch arrays, biomimetic mosquito needles that penetrate through the dura of birds, and porous electrodes with enhanced surface area. The rapid 3D printing and wafer-scale methods described here will enable efficient device fabrication and new studies examining the relationship between electrode geometry and electrode performance. Applications include small animal models, nerve interfaces, retinal implants, and other devices requiring compact, high-density 3D electrodes.

New technologies are required for high-density neural recording in animal studies and human clinical devices. While several promising electrode technologies have emerged in recent years, the devices with the highest impact must be manufactured efficiently. Typically, this means devices built in a cleanroom via thin-film processes developed for the integrated circuit industry. Examples of electrodes in this class include polymer electrodes designed at Lawrence Livermore Labs[1] and Neuralink Inc.[2], silicon electrodes developed in Michigan and Neuro-Nexus Technologies, Inc.[3], and the transformative, high-density silicon electrodes designed by Neuropixel[4] and the Italian Institute of Technology[5]. While cleanroom fabrication methods provide the required miniaturization and scalability, thin-film devices are, by nature, planar. However, new studies report the possibility of folding thin-film electrodes for 3D recording in vitro[6]. Most three-dimensional (3D) electrode structures, such as microwire arrays in a "bed of nails" design, have historically been assembled by hand. The Utah array is currently the most widely used 3D electrode array. Fabrication of this array involves mechanical silicon cutting or dicing. This step requires

rigid backing, and the cutting tool limits the minimum spacing between electrodes[7]. As a result, Utah arrays are too large to be used in small animals such as mice and songbirds, and their relatively large shanks also limit chronic performance due to foreign body tissue responses. The Plexon N-Form (3D) array has similar physical size constraints, prohibiting many applications in small animals, nerves, or retinas.

3D-printed electrodes provide a new alternative to current electrode designs. Recent devices developed at Carnegie Mellon University have demonstrated the concept of 3D-printed bed-of-nails electrodes using an aerosol jet conformal printing method[8]. Though groundbreaking, the devices are limited by low resolution (10 μm) in the aerosol jet process.

This work describes a 3D electrode array fabricated using two-photon lithography and thin-film fabrication processes. Our goal was to create a fabrication process for a new electrode whose form factor is amenable to chronic recording in small animals. However, in this report, we do not perform any chronic implants. While multielectrode

[1]Phil and Penny Knight Campus for Accelerating Scientific Impact, University of Oregon, Eugene, OR, USA. [2]Department of Bioengineering, The University of Texas at Dallas, Richardson, TX, USA. [3]These authors contributed equally: Morgan A. Brown, Kara M. Zappitelli. ✉e-mail: timg@uoregon.edu

arrays fabricated on silicon via two-photon lithography were reported over a decade ago, the fabrication steps utilized either nanoimprint or proximity-mask photolithography, limiting 3D structures to a maximum height of a few micrometers. Furthermore, the previous devices were exclusively fabricated on rigid glass or Si substrates[9–11]. Here, we report a fabrication process yielding high aspect-ratio devices (>10:1) integrated on flexible polyimide or parylene C films in a form suitable for chronically implanted devices. The 3D printing process described here is adaptable to various designs, allowing the creation of distinct height profiles along the electrode array and different electrode shapes – fully customizable electrode arrays that conform to specific anatomical features of the brain.

## Results

### High-speed custom 3D direct laser writer

Two-photon lithography is a 3D printing method that uses femtosecond pulses of infrared light to polymerize an ultraviolet photoresist at the focal point of a high-numerical-aperture lens. By changing the position of the focal point within the liquid photoresist, complex polymer shapes can be written at micron resolution. We recently described an open-source 3D printing system that uses a resonant scan mirror to increase printing speeds by 1–2 orders of magnitude relative to most galvanometer-based printers[12]. In this system, a design that fills a cubic millimeter of space at near-micron resolution takes approximately one minute to print. The printer incorporates fluorescence imaging and reflected-light-sensing pathways that provide real-time information about the degree of polymer crosslinking and surface localization. During the print process, surface localization within a

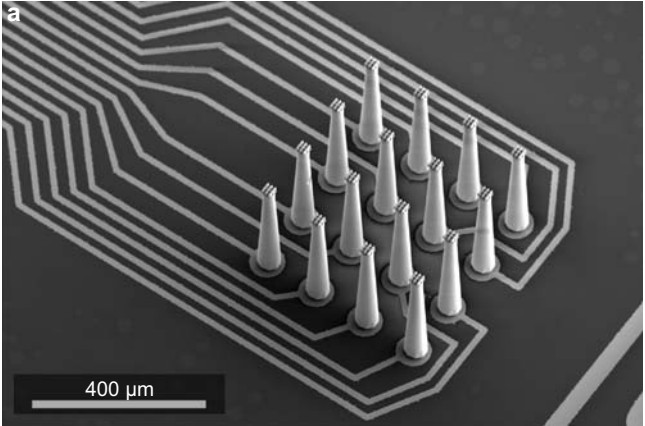

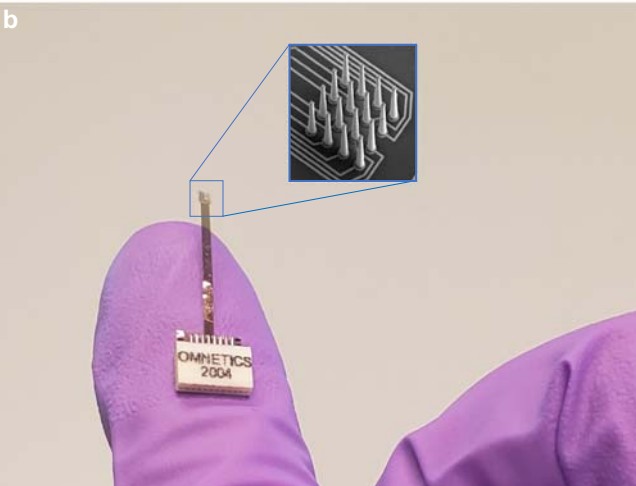

**Fig. 1 | 16-channel electrode array. a** SEM micrograph showing traces and 3D-printed electrodes fabricated via direct laser writing. **b** Assembled device showing polyimide flex cable, finger for scale.

micron is critical to achieving strong adhesion between the 3D print and the substrate. The printer achieves this by measuring the fluorescence of the photoresist at the substrate interface. For all devices reported here, we used the commercially available photoresist OrmoComp® (Micro Resist Technology), a glass-like, biocompatible member of the ORMOCER® family[13]. We provide specific details of the photoresist blend in Methods.

### 3D-printed arrays for neural recording

We use two-photon lithography to 3D print non-conductive structures on thin films. The 3D structures are later coated with platinum for electrical conductivity to make functional electrode arrays. The first prototype device was a 16-channel array of 350 μm tall electrodes with a 20 μm diameter at the recording tip, Fig. 1. The electrodes are spaced at 90 μm, and the device includes a hybrid polyimide/Parylene-C flex cable running from the electrodes to an external connector (as shown in Fig. 1a). Individual shanks in this prototype device were spaced too closely to be implanted in our test organism, the zebra finch songbird. This was due to a "bed of nails" effect upon insertion. After describing this first electrode, we show devices made implantable by either increasing the spacing of electrodes or preparing sharper tips.

The fabrication diagram in Fig. 2 outlines a simplified device geometry, which does not include the 16 mm long polyimide ribbon cable shown in Fig. 1. The first five steps in Fig. 2 are standard wafer-scale thin-film processes. The process is described in more detail in the Methods section. Briefly, a Chromium (Cr) sacrificial layer is deposited and patterned to form the outline of the traces and electrode sites. Cr is used for metal trace patterning instead of standard lift-off photo-resists since the development of the 3D prints also dissolves the standard photoresists we tried. To integrate the 3D electrode structures in step 6, the wafer is transferred to the 3D printer, and a liquid photopolymer is applied. The microscope objective is immersed in liquid photopolymer, and the 3D shapes are written directly with the laser from an input CAD file. Each 16-channel array of polymer spikes is printed in approximately 10 min using an integrated surface finding and substrate positioning feature. Total print time can be accelerated if more than one spike is printed at a time, but this was not explored in this paper. Following the printing process, the entire wafer is metalized via a non-directional sputter deposition process. The sputtering process metalizes both the traces and polymer-printed structures simultaneously. Lift-off of the Cr layer is performed to define the traces and electrodes.

The wafer is coated with parylene C via vacuum deposition, a standard insulation process used for electrodes[14] to insulate the metal traces and metalized spikes. A small region of the parylene C insulation layer is removed at the tip of each electrode to expose the underlying platinum to create a recording surface. In prior reports, this has been achieved using a mask and UV laser process[14] or a focused ion beam (FIB)[15]. The FIB process is slow and costly, and single-photon laser ablation with a mask provides little flexibility for opening specific locations in complex three-dimensional structures. In our fabrication process, parylene C removal is achieved via femtosecond laser milling using a high-pulse, energy-amplified fiber laser (Monaco by Coherent, max 40 μJ). To do this, we utilize the same resonant scanning two-photon microscope that forms the basis of the 3D printer after first switching laser sources from the 80 MHz polymerizing laser to a high pulse energy laser. Ablation involves raster scanning a 2–3-micron thick volume that includes all electrode tips (see Methods), removing material at micron resolution. Figure 3 illustrates the tip-opening process. The exposed metal is evident in Fig. 3d, e shows a scanning electron microscope (SEM) image with false color based on energy-dispersive X-ray (EDX) chemical analysis. The blue false color on the exposed tip indicates a surface presence of the element platinum.

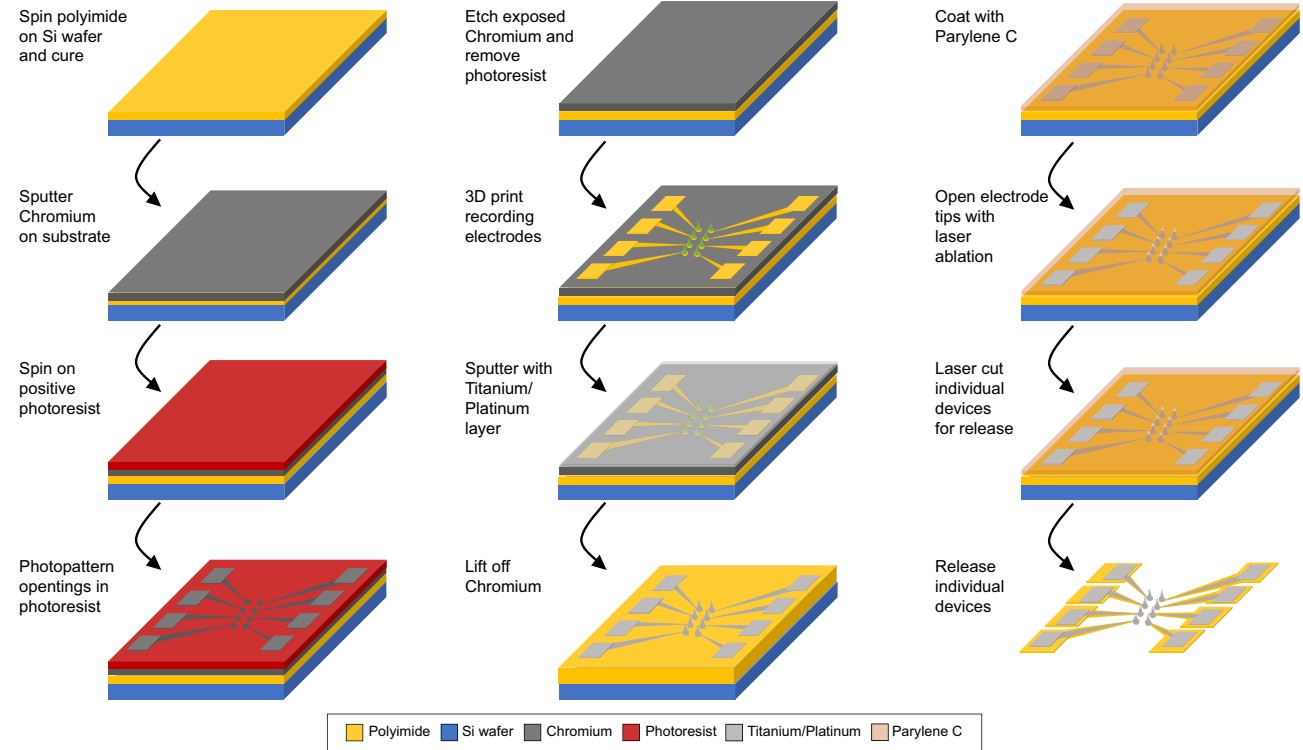

**Fig. 2 | Fabrication process.** A Chromium sacrificial layer is patterned over a polyimide coated substrate to define both the traces and electrode sites. Electrode structures are 3D printed, developed, and sputtered with a Ti/Pt layer. The Chromium sacrificial layer is then removed, leaving the Platinum prints and traces. Next, a parylene-C layer is added over the entire device for insulation. In the final steps, this insulating layer is selectively removed from the tips of the electrodes and the device perimeter is laser cut using a femtosecond laser (Monaco by Coherent). Devices are released from the wafer by soaking in heated DI water.

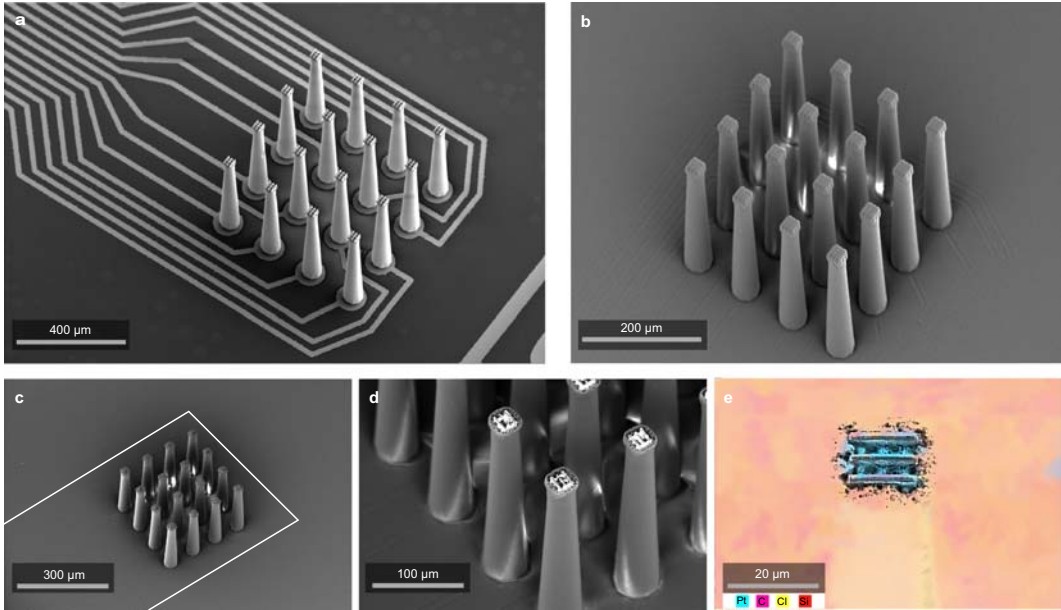

**Fig. 3 | Fabrication of a 16-channel electrode for neural recording. a** Platinum coated array on the wafer before insulation. **b** Array coated in 3 μm of parylene C insulation. **c**–**e** Images taken post tip exposure. The connecting traces are outlined with a dashed line (**c**). The exposed platinum is evident in **d**, and **e** shows an EDX of the exposed tip with platinum in blue.

Thanks to the a-thermal process of femtosecond milling, this tip-ablation process does not leave significant re-deposited material on the electrodes. No additional cleaning steps were used in Fig. 3d.

Figure 4 illustrates a cross-section of the device and impedance spectra from 16 electrodes on a single test device. The mean impedance of the 16 electrodes is 200 kOhm at 1 kHz. Note that in these initial tests, a small "log pile" structure is 3D printed at the top of each electrode (Fig. 3e). This structure creates an additional surface area at the recording. After laser ablation, the exposed electrode surface areas yielded acceptable impedance values, though future

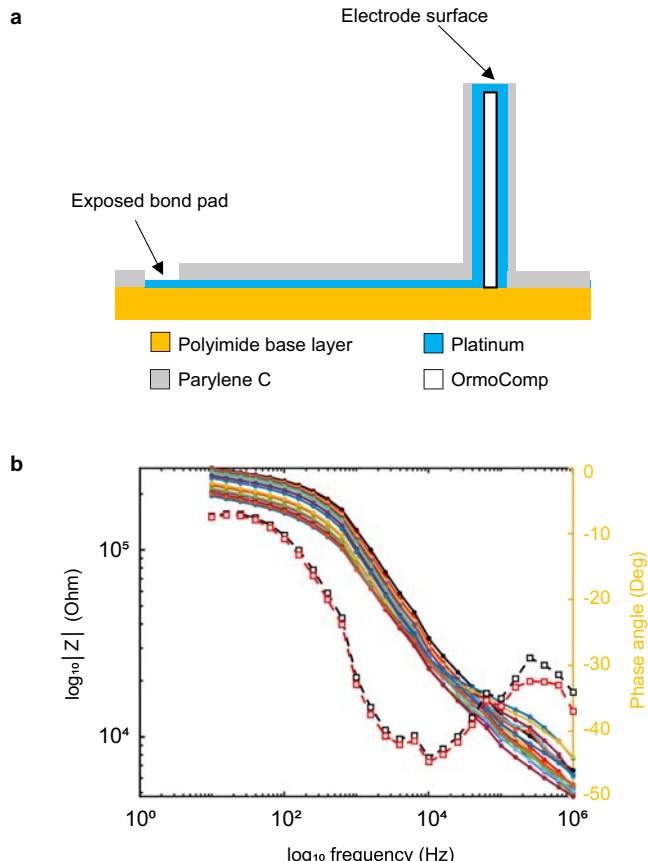

**Fig. 4 | Recording surface and impedance. a** Prototype electrode cross-section. **b** Impedance spectrum of all 16-channels from a device (filled circles) and example phase traces (open squares).

fabrication runs could include a subsequent IrOx electrodeposition step to decrease the impedance[16].

As a final fabrication step, the devices are singulated by scanning the device outline with the high pulse energy laser to cut through the parylene C and polyimide. Using anisotropic conductive film (ACF), the flex cables are bonded to an Omnetics connector allowing for direct integration with Intan and Open Ephys recording systems.

The fabrication process described here provides a general recipe for high aspect-ratio 3D-printed electrodes integrated on thin-film substrates. The automated printing and laser ablation steps are algorithmically defined, making it easy to alter electrode height profiles or shapes and customize arrays to specific experiments or brain anatomical features.

### Neural implantation and recordings

We recorded in zebra finch and mouse to test whether these devices can capture relevant physiological signals from the brain. We positioned the devices precisely with a micromanipulator, holding and applying load directly behind the array with a suction pipette (Fig. 5a). Once implanted in the zebra finch brain, the devices picked up high-SNR spikes on multiple channels (Fig. 5b), showing that the prototype can capture neuronal populations at the single-unit level. Figure 5b shows example spikes that exceed an SNR of 8 on their respective channels from the implantation shown in Fig. 5a. We next recorded local field potentials (LFPs) in the olfactory bulb of awake, freely breathing mice to test whether our devices can capture neural correlates of ethologically relevant behavior. Figure 5c shows an example raw recording from the olfactory bulb of a mouse and two isolated units. Spikes and LFPs in the olfactory bulb faithfully follow the

breathing rhythm[17–19]. By aligning LFPs with the breathing signal, we show that LFP rhythms recorded from our devices faithfully recapitulate the aperiodic breathing rhythm of awake mice (Fig. 5d). These data demonstrate that our devices can cleanly capture relevant physiological signals at high temporal resolution. Future implementations will enable denser, chronic recordings, even in hard-to-reach brain areas.

### Biomimetic geometries and insertion tests

Based on Euler's buckling calculations for the 1 GPa modulus of the photoresist OrmoComp® (Micro Resist Technology), individual electrodes with 20 μm flat tipped geometry will withstand a critical force of 1–3 mN, which is more than an order of magnitude above the penetration requirement for similar electrode shapes in another study that inserted electrodes in mice brains with dura removed −0.17 mN[20]. However, our preliminary surgeries determined that multielectrode arrays composed of flat-topped 20 μm electrodes at a pitch of 90 microns could not be inserted in songbird brains due to dimpling of the brain surface—the "bed-of-nails" effect. We found that the same 20 μm electrodes at 300 μm pitch were easily inserted in songbird brains with the dura removed.

To reduce tissue insertion forces, we next developed a prototype of an electrode that mimics a natural geometry[14]. The mosquito proboscis reduces insertion force while resisting buckling thanks to a tip geometry shown in Fig. 6b. Taking advantage of the high resolution of the 3D print process; we printed test structures with sharp spikes resembling the point of the mosquito needle (Fig. 6a). Test structures were composed of 350 μm tall shanks in a 4 × 4 grid at 150 micron pitch. Another study found that 15 micron wires with a 24-degree beveled tip showed comparable pia insertion forces to flat-topped wires. Still, electropolished wires with a 10 nM tip radius revealed insertion forces that were decreased by an order of magnitude or more relative to flat wires[20]. We could not measure insertion forces during these tests, but the devices could be readily inserted in songbird brains with the dura removed. Surprisingly, we also found that the mosquito-needle prototypes could be inserted in songbird brains without removing the dura. An implant that avoids durectomy will be faster and reduce the risks of swelling and damage to the brain surface. The mosquito needle provides proof of concept. The rapid and low-cost printing process can efficiently explore a range of electrode geometries in search of an optimal shape for tip insertion.

### Porous stimulating electrodes

In addition to neural recording, microfabricated electrodes produced using thin-film lithography have been proposed for high-channel-count neuromodulation. Applications include bioelectric medicine via the stimulation of peripheral nerves[21,22], visual prosthesis via retinal interfaces[23], and cortical interfaces to provide neuromodulation for mental disorders or enhance stroke recovery[24,25]. Multiple companies are now pursuing neural interfacing through microfabricated flexible electrode arrays, and these projects include future applications in neural stimulation[2,26,27].

Traditional microfabricated electrode arrays are, by nature, planar. One way to improve their stimulation performance is to create 3D structures raised above the electrode surface, allowing for a more intimate connection with the target tissue. This can concentrate charge delivery to target neurons, improving both stimulating thresholds and specificity. This concept was investigated in a silicon retinal interface with stimulating electrodes composed of vertical pillars, 10 μm in diameter and 65 μm tall, that interfaced directly with the inner nuclear layer of cells in the retina[28]. This intimate contact facilitates concentrated charge injection for retinal prosthesis and lowers stimulating voltage—factors that could lead to a higher-resolution visual prosthesis with lower power requirements[29].

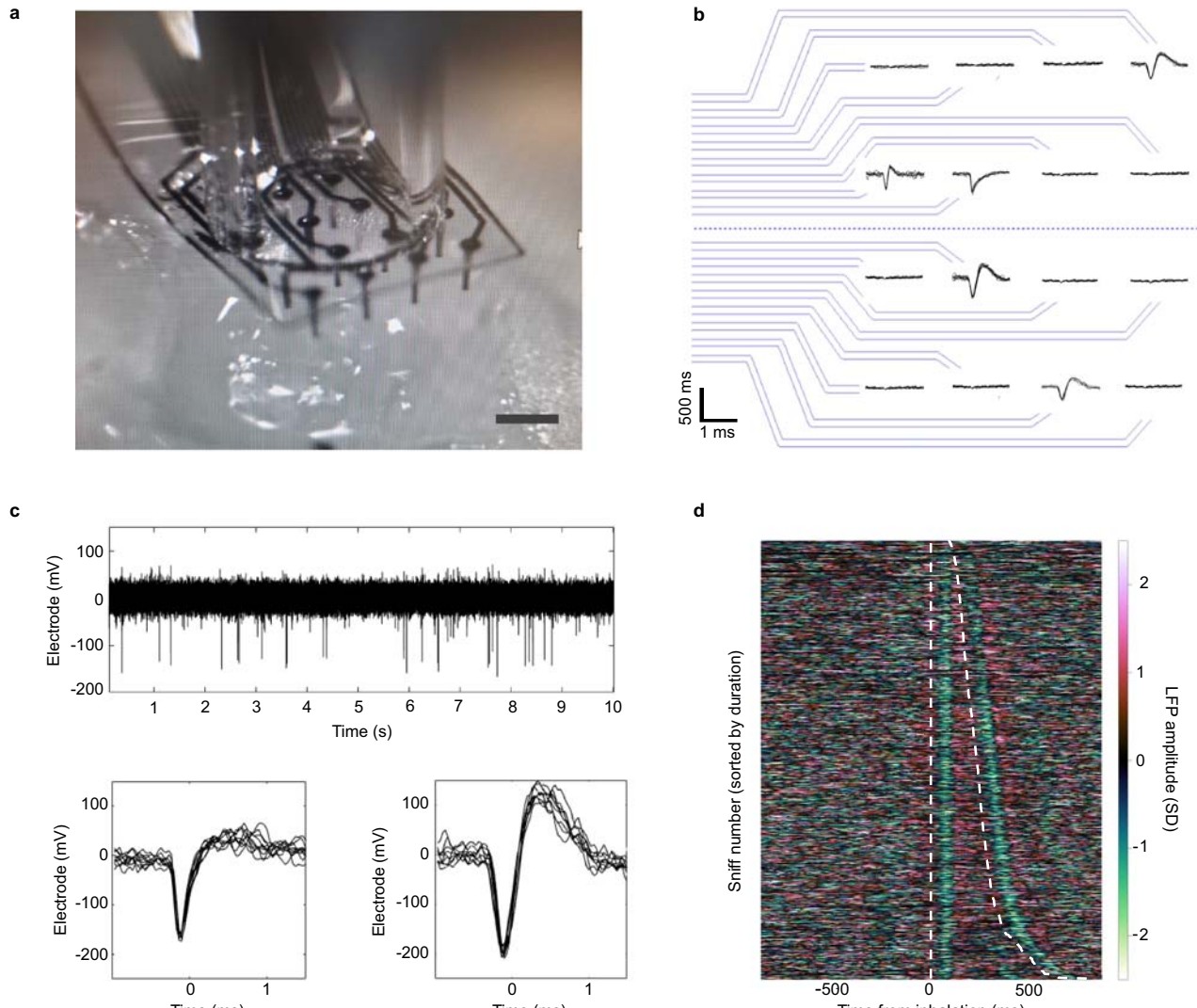

**Fig. 5 | Neural recordings in diverse species. a** MEA imaged just prior to insertion in zebra finch craniotomy (200 μm scale bar). **b** 16-channel device schematic with spike waveforms from zebra finch recording prep shown in **a**. Waveforms shown on their respective channels where SNR > 8 was observed; noise trace shown on channels, which did not meet this criterion. **c** Spike waveforms recorded in the olfactory bulb of mice. Multiple spike waveforms can be extracted from each electrode (Lower panels). **d** Inhalation (white dashed line) synced local field potential (LFP) recorded in the olfactory bulb of mice.

While 65 μm tall vertical pillars on rigid silicon are an appropriate form factor for small, localized implants, devices that interface with a more significant fraction of the retina or other curved surfaces will require the integration of stimulating pillars on a flexible substrate. Similarly, some applications, such as cortical stimulation, may require higher aspect-ratio stimulating electrodes for therapeutic charge delivery. We anticipate that the thin-film 3D electrodes described in this project could be adapted to a stimulating electrode geometry. As a first step toward that goal, we developed a process to create macroporous 3D electrodes. These devices were initially fabricated on silicon wafers for easy electrochemical testing.

Figure 7 illustrates the fabrication process for the porous electrodes on a Si substrate. The metal lift-off process described in Fig. 2 involved fabricating electrical traces in the same sputtering step that coats the 3D prints with metal. In contrast, this process involves pre-established traces later connected to raised 3D metal surfaces in a second metal sputtering step. We printed structures with pore cross-sections ranging from 40 μm² to 400 μm², solid prints, and planar electrodes lacking 3D prints for controls. Sputtering at elevated pressure was used to reduce directionality, resulting in interior metalization of the porous structures.

Figure 8 shows SEM micrographs of the structures. These prototypes feature metalization throughout the interior of a porous 3D shape. Figure 8c, d shows the extent of the internal metalization of a print both before and after FIB sectioning. A complete device can be seen in Fig. 8a. For the solid pyramids, we found a ~2× increase in charge storage capacity relative to a flat 2D electrode pad (Fig. 8b). However, the solid and large-pore pyramids showed similar cyclic voltammetry curves even though the surface area of the large-pore pyramid is 2x the surface area of the solid pyramid. While the metalization in Fig. 8d appears to be conformal on the interior of the 3D electrode, we cannot rule out that the interior is inadequately coated, and further tests will be needed with longer platinum deposition runs. Alternatively, the interior of the porous electrode may not be completely accessible to the process of electrochemical charge injection in saline due to time constants of diffusion or other factors. In future studies, a given electrode geometry can be printed at multiple scales to systematically examine the impact of pore dimensions on charge injection capacity.

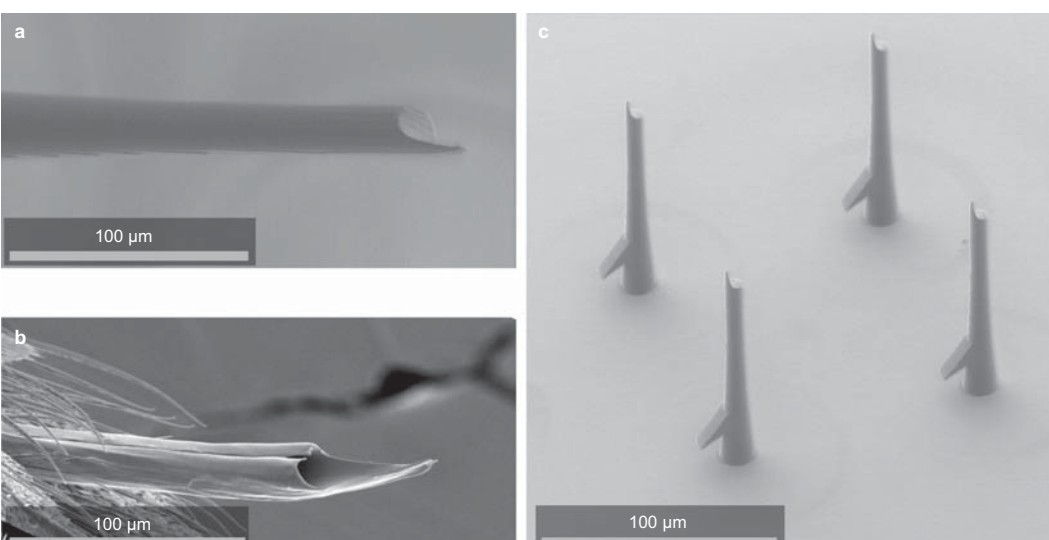

**Fig. 6 | Biomimetic electrodes. a** Biomimetic mosquito-needle tips incorporate a sharp tip for reduced insertion force, as observed from an actual mosquito (**b**, adapted from Wikimedia Commons). **c** These probes can be inserted through the dura of birds.

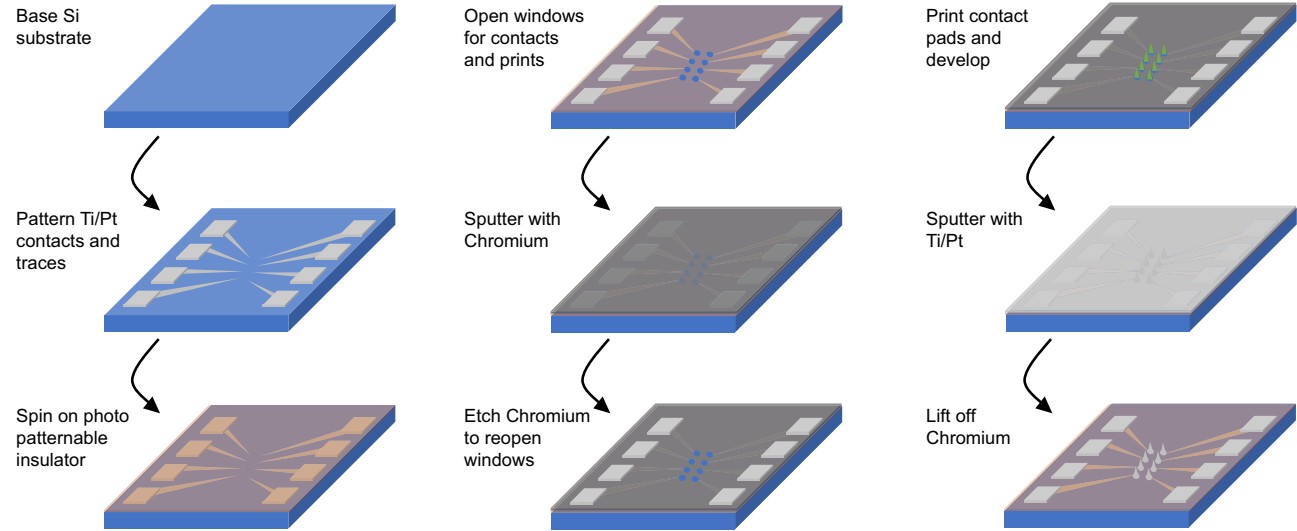

**Fig. 7 | Fabrication process for Si supported devices.** The metal contact pads and traces are patterned directly on the Si wafer. The traces are then insulated with SU-8 and coated in a Cr sacrificial layer. The SU-8 and Cr layers are both patterned, opening windows for prints and the contact pads. Electrode structures are 3D printed, metalized, and then the Cr is etched to lift-off unwanted metal.

## Discussion

We have established a process to create 3D electrodes with flexible geometry fabricated at micron resolution. This emphasis on maximizing resolution in 3D-printed electrodes is motivated by multiple factors. First, the high resolution and design flexibility in two-photon lithography makes exploring a wide range of novel electrode shapes possible. Biomimetic mosquito needles penetrate through the dura, barbs that allow electrodes to anchor within the tissue, and porous electrodes are examples of shapes shown here. The 3D printing process enables unique individual electrode geometries and the customization of electrode length profiles to match specific brain regions' curvature or depth profiles. Another potential benefit of high-resolution 3D printing is that it may reduce electrode cross-sections, leading to chronic neural recordings with a higher signal-to-noise ratio (SNR). For electrodes that are significantly larger in cross-section than neuron cell bodies, a reactive tissue response encapsulates electrodes and damages cells at distances up to 100 μm from the implant[30–34].

Action potential amplitude decays rapidly with distance from an electrode[35–37]. As a result, adverse tissue response is a particularly severe problem for small animal studies where scar encapsulation can prohibit single neuron resolution recordings from the densely packed regions of interest. The two-photon lithography employed here may allow the fabrication of microelectrodes with dimensions well below the 20 μm limit that is thought to evade much of the brain's immune response[38–40]. To realize this vision, it will be necessary to increase the stiffness of the printed structures to create finer electrode shanks that can insert without buckling and to develop laser tip-ablation strategies that preserve sharp geometries, such as the mosquito-needle prototypes. Increased stiffness will be required to create a device complementing the Utah array for human or large animal chronic applications in Cortex. For this application, electrodes should be at least 1 mm long. Electrodes of this length can be printed with the method described here but will likely require enhancements in stiffness to avoid buckling during insertion. Current directions in the

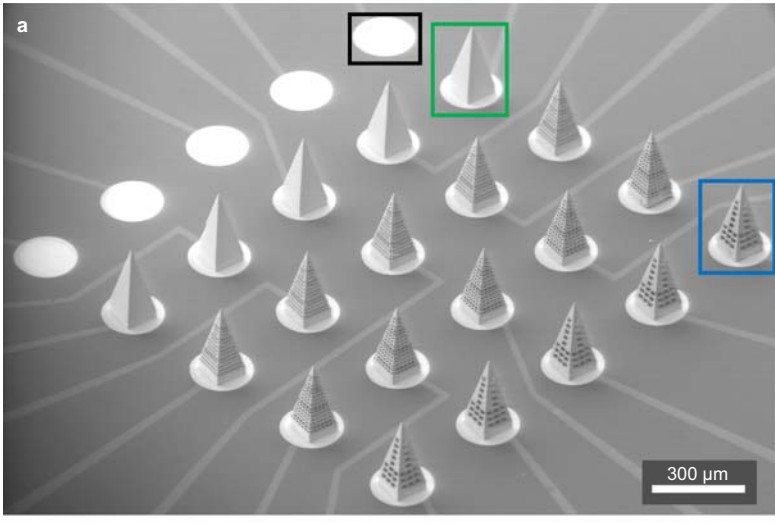

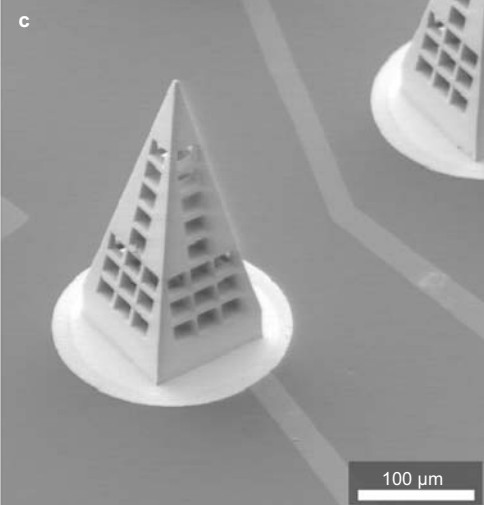

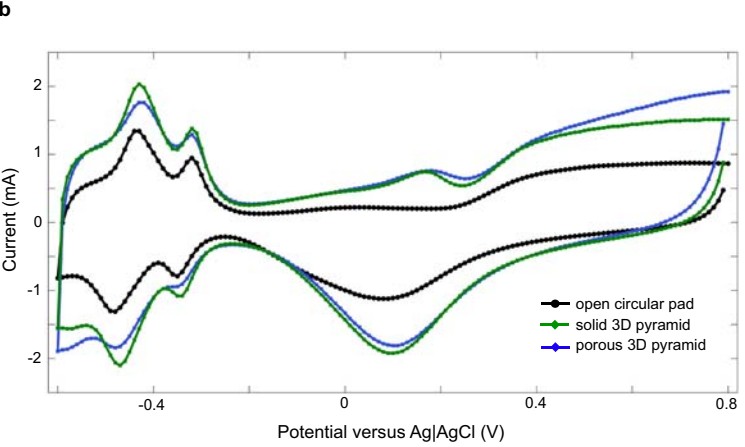

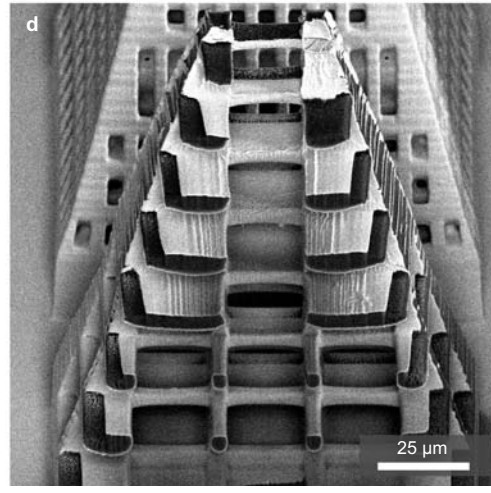

**Fig. 8 | Stimulating electrodes. a** Metalized electrode array on a Si substrate with SU-8 isolation of traces only. **b** Example CV curves gathered at 50 mV/s reveal increased charge storage capacity for 3D pyramids (solid or porous [20 μm pores]) relative to flat circular pads. **c**, **d** SEM images of a print before and after FIB sectioning.

project are exploring the use of amorphous silicon carbide as an alternative to the parylene-C encapsulation layer, thereby increasing electrode stiffness.

The high resolution of two-photon lithography could enable the fabrication of high-channel-count electrode arrays, resulting in a more significant number of electrodes per unit area within the brain or retina. These devices represent a region of electrode configuration space previously unoccupied, namely the potential for high-channel-count 3D electrode arrays with features definable at micron resolution.

### Small animal model electrodes

We anticipate initial applications in small animal models where 3D electrode arrays could be fabricated to conform to specific spatial profiles within target brain regions. In addition to chronic implants in fixed locations, integrated flex cables allow these devices to be mounted on micro-drives to sample multiple depths serially. In song-birds and other animal models, the quality of single-unit isolation decays after implantation, making it challenging to study the neural mechanisms of learning. The only known way to significantly improve signals over the long run is to reduce electrode scale. While carbon fiber electrode arrays have demonstrated stable recordings over long time scales, yield is low, and the fabrication process is not scalable[41]. Thin-film silicon carbide ultramicroelectrodes provide a

scalable alternative that will be investigated further[42], and polymer electrodes using insertion shuttles show promise for increasing signal longevity[43]. Until the progressive signal loss is resolved, the highest signal-to-noise ratio recordings will be achieved by moving multi-electrode arrays into fresh brain tissue with a micro-drive. Micro-drives that advance an electrode with a screw or small motor have been the cornerstone of electrophysiological studies in songbirds and mice for decades[44], and high-channel-count silicon probes have increased the yield of these experiments[45]. Still, for a single silicon shank with many electrodes, only the electrodes near the probe's tip are moved into fresh tissue when a micro-drive is advanced. For the 3D electrodes described here, every electrode contact will move into new tissue when the micro-drive is advanced. We anticipate that high-channel-count 3D electrodes on micro-drives will yield a more significant number of single-unit recordings, extend the operational period of animal experiments, and provide the unique ability to sample 3D volumes in behaving animals.

### Porous stimulating electrodes

We anticipate multiple benefits from 3D and porous stimulating electrodes. At the simplest level, the protruding surfaces of 3D stimulating electrodes will provide better electrical contact between the electrode and neural tissue, with potential applications in cortical micro-ECoG recording and stimulation, as well as peripheral nerve interfacing[29].

3D-printed macro-pores can also increase the surface area of the electrodes while maintaining the same overall displaced tissue volume. For example, the solid pyramids in Fig. 8a have a surface area of 0.076 mm$^2$. In comparison, the pyramids with the smallest pores have a total surface area of 0.391 mm$^2$, providing a five-fold increase in contact between the stimulating surface and the tissue. The benefit of this porosity remains to be determined for chronic implants where tissue ingrowth leads to increased access resistance within the pores. While tissue ingrowth may be a limiting factor for charge injection, it could stabilize the porous neural interface against micromotion and reduce fibrotic encapsulation. Ingrowth of neural processes could also be encouraged to promote "neurotrophic electrodes" with the potential for stable multi-unit recordings and reduced thresholds for stimulation of ingrown axons or dendrites[46–48]. Recent studies have demonstrated vascular integration of porous ECoG arrays implanted at the brain's surface in mice[49]. We speculate that similar vascular ingrowth could occur in porous, 3D-printed microelectrodes and could improve device longevity by providing stabilization against migration and micromotion.

The fabrication method for 3D-printed electrode arrays presented here is robust, wafer scale, and fully compatible with standard Si and flexible polyimide device fabrication processes. Using high-resolution 3D laser writing, a wide range of unique electrode shapes can be fabricated, from biomimetic needles to porous electrodes. These devices will enable volumetric recordings at a spatial resolution not achievable with current 3D microelectrode devices. The fabrication methods described here can be adopted at many research universities for users with access to two-photon lithography, laser ablation, photo-lithography, and sputtering tools. If dissemination grants or industry partners support this work, the technique could provide new tools for a range of users across neuroscience and neural engineering and in human applications such as visual prosthesis or nerve interfaces where high-density recording and stimulation are required in a small form factor.

## Methods

### 3D-printed electrodes

Three-dimensional electrode structures are printed using a two-photon 3D printer in a process known as direct laser writing. This printer was described previously[12]. Electrode shapes are designed in standard 3D CAD software and uploaded into the printer software as STL files for print voxelization. The photopolymer is a hybrid resist based on the commercially available photoresist OrmoComp® (Micro Resist Technology), a glass-like, biocompatible member of the ORMOCER® family[13]. We add a photoinitiator (2,4,6-trimethyl benzoyl phosphine oxide (TPO), Sigma Aldrich), a stabilizing agent, (3,5-Di-tert-butyl-4-hydroxytoluene (BHT), Sigma Aldrich) and fluorescein (Sigma Aldrich) for in situ imaging during 3D printing. A 780 nm Chameleon Discovery laser with a 100 fs pulse width, 80 MHz rep rate, set to ~40 mW power, is focused through a 20x Nikon immersion lens (NA 0.7) to initiate polymerization in the photoresist. Following prints, the substrates are submerged in Ormodev (Micro Resist Technology) developer for 12 h to remove un-polymerized photoresist, followed by a rinse in isopropanol. The development process is followed by a 10-min UV cure at 395 nm (Solis-365 C at 2.8 mW/mm$^2$) to increase the overall degree of crosslinking in the polymerized resist and to enhance the mechanical stability of the structures[50].

An overview of the thin-film fabrication process can be seen in Fig. 2. Electrodes are fabricated on prime grade 75 mm Si wafers with 300 nm of thermal oxide (University Wafer). A base layer of polyimide (HD MicroSystems PI2611) is spun onto the surface and cured at 350 °C for 30 min. in a nitrogen environment to a final thickness of 6 μm. Adhesion promoter (HD Microsystems VM652) is added to the edge of the wafer before polyimide spin coating. 500 nm of Cr is then sputtered onto the polyimide surface (3 mTorr DC)

as a sacrificial layer. To define the metal traces of the electrode array, AZ-1512 photoresist (Kayaku Advanced Materials, Inc.) is spun onto the wafer surface and patterned using a Süss Microtec MJB4 mask aligner (350 W mercury arc lamp) with an exposure dose of 70 mJ/cm$^2$. All photolithography masks are written in-house on a direct write laser lithography system (Heidelberg). After development (AZ 300 MIF), room temperature Transene Chromium Etchant 1020 (ceric ammonium nitrate/nitric acid) is used to etch through the 500 nm Cr layer, forming the mask for the final traces and defining the print locations. The photoresist is removed via sonication in acetone at 37 kHz for 5 min.

Before 3D printing, the patterned wafer is cleaned with oxygen plasma for 90 s in a March plasma etcher at a pressure of 300 mTorr and RF power of 100 W to increase print adhesion. Following initial wafer alignment, surface finding and printing for each electrode in the array is automated. After printing and developing 3D electrode structures, the devices are plasma cleaned again before final metalization to increase metal adhesion to the prints. The print structures are then sputtered with Ti (15 nm)/Pt (200 nm) in an Angstrom Engineering sputter system at 3 mTorr (Ti−DC) and 10 mTorr (Pt−RF). The Cr sacrificial layer lift-off is done in Transene Chromium Etchant 1020 at 60 °C for 15 min. To effectively remove all metal flakes, wafers are transferred to multiple fresh etchant baths during the lift-off process with agitation, followed by a thorough rinse in DI water. Finally, the Omnetics connector contact pads are masked with Kapton tape, and a 3 μm thick layer of parylene C is deposited (Labcoater 4200) over the wafer.

The porous electrodes illustrated in Fig. 8 were fabricated directly on the Si wafers described in Fig. 7. Here, an initial metalization (10/50 nm Ti/Pt) layer was patterned and then electrically isolated with 500 nm of SU-8 negative photoresist (Kayaku Advanced Materials, Inc.). Openings in the SU-8 were made using a photomask and an exposure dose of 65 mJ/cm$^2$. These openings were large enough to include a region of the Ti/Pt metal traces to create electrical connections in the subsequent metal sputtering. 500 nm of Cr is then sputtered onto the polyimide surface (3 mTorr DC) as a sacrificial mask layer. AZ-1512 photoresist (Kayaku Advanced Materials, Inc.) is spun onto the wafer surface and patterned to define the print location holes in the Cr aligned with the holes in the SU-8. Patterning of this layer is done with a Süss Microtec MJB4 mask aligner (350 W mercury arc lamp) with an exposure dose of 70 mJ/cm$^2$. After development (AZ 300 MIF), room temperature Transene Chromium Etchant 1020 is used to etch through the 500 nm Cr layer, forming the openings for the final print locations. The photoresist is removed via sonication in acetone at 37 kHz for 5 min. Devices were then printed, metalized, and the chromium layer was released as described above.

### FIB milling

Focused ion beam (FIB) milling was utilized for slicing open printed structures and assessing the extent of internal metalization. A ThermoFisher Helios Hydra PFIB was used with an Ar/O$_2$ beam.

### Laser tip ablation

A 1035 nm wavelength pulsed laser (Coherent, Monaco) is used to remove the parylene C from the tips of the prints. This laser is co-aligned with the 3D printing laser (Discovery) in the above resonant scan 3D printing system. The initial alignment points on the wafer are found using the Discovery laser to avoid damage. The parylene C is then removed from the tips of all 16 electrodes in a single cut process at a 1 MHz pulsing setting. The cut raster scans a 2–3 micron thick volume, including all electrode tips. During this process, emitted ultraviolet from the ablation process is imaged with photomultiplier tubes in a standard two-photon imaging fashion. This emitted light from the laser ablation process can be used to calibrate the ablation power. This laser ablation process takes less than one minute per array.

## Device finalization and release

To release the entire device from the wafer, the 1035 nm pulsed laser is used to cut through the polyimide and parylene layers using the programmed motion of a precision translation stage. The wafer is then placed in warm water to release the individual devices. Finally, Omnetics connectors are attached to the device pads via anisotropic conductive film (3 M, ACF 7371).

## Electrochemical measurements

Cyclic voltammetry (CV) and electrochemical impedance spectroscopy (EIS) data were collected using a high surface area Pt counter electrode and an Ag/AgCl reference electrode on a Gamry Reference 600 potentiostat. Measurements were conducted in phosphate-buffered saline (pH ~7.2) consisting of 0.126 M NaCl, 0.081 M $Na_2HPO_4$, and 0.022 M $NaH_2PO_4$. Before measurement, the electrolyte solution was sparged with He gas for ~30 min to remove dissolved $O_2$. CV curves were cycled at 50 mV/s until differences between subsequent scans were no longer observed. Additional complete device checks were performed via Open Ephys using Intan chips.

## Animals

All experimental procedures were approved by the Institutional Animal Care and Use Committee (IACUC) at the University of Oregon and comply with the National Institutes of Health Guide to the Care and Use of Laboratory Animals. Adult zebra finches (*Taeniopygia guttata*; >120 days post-hatch) and C57BL/6J mice (2–14 months old) were obtained from the Terrestrial Animal Care Services (TeACS) at the University of Oregon.

## Surgeries

Zebra finches were anesthetized using isoflurane gas anesthesia (1.5% in oxygen). A small craniotomy was performed above the songbird analog of the motor cortex (HVC). Acute spontaneous recordings were performed under anesthesia.

Mice were anesthetized using isoflurane gas anesthesia, and a head plate was added. A thermistor was implanted between the mice's nasal bone and inner nasal epithelium to measure respiration. A small craniotomy was performed above one of the olfactory bulbs, contralateral to the side of thermistor implantation. The reference electrode was implanted in the cerebellum. At the end of the procedure, the craniotomy was covered with a biocompatible silicone elastomer sealant (Kwik-cast, WPI). The mice were given 3 d after surgery for recovery.

## Recordings

Probes were inserted via a custom fixture, and all recordings were taken via the OpenEphys software package. The data were acquired using a 128-channel data acquisition system (RHD2000; Intan Technologies) at a 30 kHz sampling frequency. We record sniffing using intranasally implanted thermistors (TE Sensor Solutions, #GAG22K7MCD419). These thermistors are attached to pins (Assmann WSW Components, #A-MCK-80030) and into the analog input of the RHD2000.

## Data analysis

Analysis of spikes and sniffing were performed in MATLAB. Electrophysiological data were analyzed via Kilosort and Phy2. Inhalation and exhalation times were extracted by finding peaks and troughs in the temperature signal after smoothing with a 25 ms moving window. Sniffs with a duration less than the 5th and greater than the 95th percentile were excluded from the analysis.

## Data availability

The electrophysiological traces reported in this paper are available upon request.

## Code availability

The custom software for 3D printing has been described previously and is available at https://github.com/gardner-lab/printimage.

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

## Acknowledgements

We thank the University of Oregon Technical Sciences Administration staff for assistance with hardware design, and the facilities and staff at the Center for Advanced Materials Characterization in Oregon. This work was supported by the Phil and Penny Knight Campus for Accelerating Scientific Impact and by the University of Oregon Initiate of Neuroscience, the NIH fellowship grant F32MH118724 (M.A.B.), and NIH grants R01NS104925 and R01NS118424 (T.J.G., co-PD/PI).

## Author contributions

Study design (M.A.B., K.Z., T.G.), experiments and data analysis (M.A.B., K.Z., L.S., M.B., R.C.Y., V.B., D.J.M.), valuable discussion (V.B., D.J.M., M.C.S., S.F.C., T.J.G.), manuscript preparation (M.A.B., K.Z., M.B., M.C.S., T.J.G.), and funding acquisition (M.A.B., T.J.G.).

## Competing interests

The authors declare the following competing interests. Patent applicant: University of Oregon. Name of the inventor (s): Timothy J. Gardner, Morgan A. Brown, Kara M. Zappitelli. Application number: UO DIS-20-042; KS Ref. No. 1505-105149-01. Status of application, Pending. The specific aspect of the manuscript covered in the patent application: Methods and systems for fabricating 3D electronic devices with 3d-printed electronic components. There are no other competing interests.
