## [Peer Review File · Nature Communications]

REVIEWER COMMENTS

Reviewer #1 (Remarks to the Author):

The manuscript “Direct laser writing of 3D electrodes on flexible substrate” reports a 3D electrode array fabricated using 3D printing techniques combined with silicon thin-film processes. While there have been several 3D printed electrodes, the proposed electrode shows high aspect ratio electrodes on a flexible substrate. The manuscript provides valuable information for researchers and shows the possibility of 3D-printed electrodes. I find the manuscript to be compelling and well-supported by the in-vivo experiments. The authors can consider some of this reviewer's comments and questions to improve the manuscript before publication.

1. In Figure 2, a Cr layer was deposited and patterned. It is required to explain the role of Cr layer in the manuscript. Also, it is recommended to describe the overall fabrication process in detail.
2. In Figure 4, it seems that another layer was electrodeposited on the platinum electrode. The material of the electrodeposited layer should be specified.
3. It seems that the electrode array used in the in-vivo experiment (Fig. 5(a)) is different from the one in Fig. 4. Please specify the dimensions of the probe array used in the experiment.
4. The authors showed the mean spike waveforms in Fig. 5(b). It is required to provide raw spike waveforms in each channel as shown in Fig. 5(c).
5. The authors claimed that they adapted natural geometry to reduce tissue insertion forces. Please estimate the reduced force in the manuscript.
6. The charge injection capacity of the pyramid electrode with pores was similar to the solid pyramid even though the larger surface area. It is required to discuss the reasons and possible solutions to enhance charge injection capacity.

Reviewer #2 (Remarks to the Author):

These authors report fabrication and testing of a very small, low weight, 3D printable electrode array for recording from small animal subjects, in this report birds and mice. The manuscript is clearly written and very well illustrated. To this reviewers' knowledge, the devices are novel.

First comment: It would be much easier on the reviewers if pages were numbered. Authors and editors please take note.

I have mild reservations about reporting the devices at this stage of development. The manuscript's opening sentence reads "New technologies are required for high-density chronic neural recording in animal studies". Certainly true, but no attempt to test the devices in a chronic implant is included in this report. Further, the authors cite multiple changes that are likely required to make a chronically stable device. The very short depth for recording means only superficial cortex is accessible without much larger pillars with greater spacing. The stiffness of the polymer becomes challenging. Some perspective on this issue would be beneficial to the readers.

While the equipment for fabrication is indeed open source, few institutions will have the two photon microscope with both high energy pulses for ablation and low energy for polymerization. If these devices can not be made for sale, it is unlikely that significant penetration will arise. Any potential user would be happy to hear plans for availability.

Many notes for completeness of information arose during the reading.

Introduction: ref 4, Neuropixels is plural.

Results: The use of fluorescence for achieving micron resolution should have a method description or a reference to the technique.

3D printing of arrays: Reading would be more efficient with a one sentence statement that the non-conducting polymer structure is printed, then metalization is added later.

On the following page, nothing is said about laser ablation to expose the contacts. These are much larger areas than the pillar tips, and I assume a good deal of material splatter will result. Is this incorrect or is other mitigation used.

Biomimetic Geometries and insertion

The reference to insertion forces as 0.17 mN should have a subject for insertion, mouse or bird, through dura, or after durotomy.

"We found that the same 20 μm electrodes at 300 μm pitch were easily inserted in songbird brains." Was this through dura or after dura modification?

Porous stimulating electrodes

1. The authors assertion: "stimulation require higher voltages leading to accelerated delamination and degradation of stimulating electrodes" is a statement without context. The references are to old literature for a week of continuous stimulation at high voltages with the metric being visible brain

damage. For short bursts of 10-50 uA, typical for research neuroscience reports, the electrode damage and tissue damage issues are unrelated to these suicide experimental reports.

2. The structures in figure 9b. show that porosity has no significant advantage and added complexity of porous pyramids does give useful improvement. In addition, these slow scan current injection capacity curves probably do not reflect current inject performance for the fast pulse (100-200 us) widths used in research neurophysiology.

Adhesion of 3D printed polymers...

Comment on materials stability and mitigation of that stability is premature until the stability of the chronic implantation recordings are established. It may be the current materials far exceed the neural activity stability of the implant.

REVIEWER COMMENTS

Reviewer #1 (Remarks to the Author):

We would like to thank the reviewer for their thoughtful comments. We have attempted to address all points in the revision.

The manuscript "Direct laser writing of 3D electrodes on flexible substrate" reports a 3D electrode array fabricated using 3D printing techniques combined with silicon thin-film processes. While there have been several 3D printed electrodes, the proposed electrode shows high aspect ratio electrodes on a flexible substrate. The manuscript provides valuable information for researchers and shows the possibility of 3D-printed electrodes. I find the manuscript to be compelling and well-supported by the in-vivo experiments. The authors can consider some of this reviewer's comments and questions to improve the manuscript before publication.

1 In Figure 2, a Cr layer was deposited and patterned. It is required to explain the role of Cr layer in the manuscript. Also, it is recommended to describe the overall fabrication process in detail.

The Cr is a sacrificial layer that allows the lift-off of the Ti/Pt layer. It enables us to metalize the printed structures/traces selectively. Typically, a polymer photoresist would be used for this purpose, but the print development process melts away the patterned photoresist, disallowing this method.

We have edited the text and legend of Figure 2 to further explain the chromium layer and the fabrication methods. Detailed fabrication steps are also provided in the Methods section, which is now mentioned in the main text.

2. In Figure 4, it seems that another layer was electrodeposited on the platinum electrode. The material of the electrodeposited layer should be specified.

We corrected the error in the text and figure. There is no electrodeposition in the method.

3. It seems that the electrode array used in the in-vivo experiment (Fig. 5(a)) is different from the one in Fig. 4. Please specify the dimensions of the probe array used in the experiment.

In vivo implants in 5 were with 300 um pitch. We clarify this in the abstract and throughout the text, explaining why we worked at different spacing.

4. The authors showed the mean spike waveforms in Fig. 5(b). It is required to provide raw spike waveforms in each channel, as shown in Fig. 5(c).

The raw waveforms have been added as requested.

5. The authors claimed that they adapted natural geometry to reduce tissue insertion forces. Please estimate the reduced force in the manuscript.

We were unable to measure the small insertion forces in our setup. We have now cited expectations from prior literature and removed the claim that the insertion was easier than the flat-topped electrodes. The take-home point of that section is now that the mosquito needle tips insert through the dura in songbirds.

6. The charge injection capacity of the pyramid electrode with pores was similar to the solid pyramid even though the larger surface area. It is required to discuss the reasons and possible solutions to enhance charge injection capacity.

We have added the following note to the text:

We note that the internal surface area of the large-pore pyramid is 2X the surface area of the solid pyramid, but we did not observe a 2X increase in charge injection. While the metalization in figure 9d appears to be conformal on the interior of the 3D electrode, we cannot rule out that the interior is inadequately coated, and further tests will be needed with longer platinum deposition runs. Alternatively, the interior of the porous electrode may not be completely accessible to the process of electrochemical charge injection in saline due to time constants of diffusion or IR drops in solution. In future studies, a given electrode geometry can be printed at multiple scales to systematically examine the impact of pore dimensions on charge injection capacity. “

Reviewer #2 (Remarks to the Author):

Reviews Comments Nature Communications MS# NCOMMS-22-53003-T

We thank the reviewer for the thoughtful comments that have motivated significant improvements in the manuscript. We hope we have addressed all issues as detailed below.

These authors report fabrication and testing of a very small, low weight, 3D printable electrode array for recording from small animal subjects, in this report birds and mice. The manuscript is clearly written and very well illustrated. To this reviewers' knowledge, the devices are novel.

First comment: It would be much easier on the reviewers if pages were numbered. Authors and editors please take note.

Thanks for this – we have added page numbers.

I have mild reservations about reporting the devices at this stage of development. The manuscripts opening sentence reads “New technologies are required for high-density chronic neural recording in animal studies”. Certainly true, but no attempt to test the devices in a chronic implant is included in this report.

A good point. Our goal was to create a fabrication process for a new electrode whose form factor was amenable to chronic recording, but we have yet to demonstrate any long-term recordings and the discussion around topics of chronic longevity was premature. We have changed language in a few places to make this clear, including removing the word chronic from the opening paragraph. The new manuscript is focussed on what was done and is now much improved.

Further, the authors cite multiple changes that are likely required to make a chronically stable device. The very short depth for recording means only the superficial cortex is accessible without much larger pillars with greater spacing. The stiffness of the polymer becomes challenging. Some perspective on this issue would be beneficial to the readers.

Excellent point - we added this paragraph to the discussion:

Increased stiffness will be required to create a device complementing the Utah array for human or large animal chronic applications in Cortex. For this application, electrodes should be at least 1mm long. Electrodes of this length can be printed with the method described here but will likely require enhancements in stiffness to avoid buckling during insertion. Current directions in the project are exploring the use of amorphous silicon carbide as an alternative to the Parylene encapsulation layer, thereby increasing electrode stiffness.

While the equipment for fabrication is indeed open source, few institutions will have the two photon microscope with both high energy pulses for ablation and low energy for polymerization. If these devices can not be made for sale, it is unlikely that significant penetration will arise. Any potential user would be happy to hear plans for availability.

Dissemination is a challenge for the field. We didn't think the manuscript was the right place to discuss dissemination plans in detail, but we have added the following conclusion. "The fabrication methods described here can be adopted at many research universities for users with access to two-photon lithography, laser ablation, photolithography, and sputtering tools. If dissemination grants or industry partners support this work, the technique could provide new tools for a range of users across neuroscience and neural engineering and in human applications such as visual prosthesis or nerve interfaces where high-density recording and stimulation are required in a small form factor."

In general, we would be happy to speak with anyone about the topic of dissemination, and certainly eager to parter with people that might like to help with that.

Many notes for completeness of information arose during the reading.

Introduction: ref 4, Neuropixels is plural.

fixed

Results: The use of fluorescence for achieving micron resolution should have a method description or a reference to the technique.

The prior manuscript is now cited.

3D printing of arrays: Reading would be more efficient with a one sentence statement that the non-conducting polymer structure is printed, then metalization is added later.

This has been added.

On the following page, nothing is said about laser ablation to expose the contacts. These are much larger areas than the pillar tips, and I assume a good deal of material splatter will result. Is this incorrect or is other mitigation used.

The laser ablation section has been expanded, including the statement that we don't observe visible redeposition in the SEM images. We also added a brief section in Methods and cite that section in the main text.

Biomimetic Geometries and insertion

The reference to insertion forces as 0.17 mN should have a subject for insertion, mouse or bird, through dura, or after durotomy.

This has been added.

"We found that the same 20 μm electrodes at 300 μm pitch were easily inserted in songbird brains." Was this through dura or after dura modification?

This test was performed after dura removal. This has been added.

Porous stimulating electrodes

1. The authors assertion: "stimulation require higher voltages leading to accelerated delamination and degradation of stimulating electrodes" is a statement without context. The references are to old literature for a week of continuous stimulation at high voltages with the metric being visible brain damage. For short bursts of 10-50 μA , typical for research neuroscience reports, the electrode damage and tissue damage issues are unrelated to these suicide experimental reports.

We have removed references to the papers and removed discussion of material longevity.

2. The structures in figure 9b. show that porosity has no significant advantage and added complexity of porous pyramids does give useful improvement. In addition, these slow scan current injection capacity curves probably do not reflect current inject performance for the fast pulse (100-200 μs) widths used in research neurophysiology.

We now highlight this in the text. At this point, we don't have an adequate explanation for the reduced charge storage of the porous electrodes relative to surface area. That section now ends with the following paragraph: "However, the solid and large-pore pyramids showed similar cyclic voltammetry curves even though the surface area of the large-pore pyramid is 2x the surface area of the solid pyramid. While the metalization in figure 9d appears to be conformal on the interior of the 3D electrode, we cannot rule out that the interior is inadequately coated, and further tests will be needed with longer platinum deposition runs. Alternatively, the interior of the porous electrode may not be completely accessible to the process of electrochemical charge injection in saline due to time constants of diffusion or other factors. In future studies, a given electrode geometry can be printed at multiple scales to systematically examine the impact of pore dimensions on charge injection capacity."

Adhesion of 3D printed polymers...

Comment on materials stability and mitigation of that stability is premature until the stability of the chronic implantation recordings are established. It may be the current materials far exceed the neural activity stability of the implant.

We agree that this section was premature and out of place in the manuscript. We have deleted the discussion of materials stability and adhesion robustness from the paper. The figure showing laser pitting of surfaces was also removed since this method was not used in the paper and was part of an unnecessary discussion about longevity.

Again, we would like to thank the reviewer for the comments that significantly improved this manuscript. We hope to have better answers for some of the remaining unknowns in the future. The focus is now on the immediate results and less speculative.

REVIEWERS' COMMENTS

Reviewer #1 (Remarks to the Author):

The authors have fully addressed all the comments in my previous report, and improved the overall quality of the work. I recommend the acceptance of the manuscript for publication in Nature Communications.

Reviewer #2 (Remarks to the Author):

The authors have addressed all my concerns and I am comfortable the manuscript can be published. In my reading I found a typo on page 6 (350-mm tall shanks) is certainly meant to be 350 μ m tall shanks.